# The Effect of Acute Physical Exercise on Natural Killer Cells Populations and Cytokine Levels in Healthy Women

**DOI:** 10.3390/sports11100189

**Published:** 2023-10-02

**Authors:** Estefania Quintana-Mendias, Judith M. Rodríguez-Villalobos, Argel Gastelum-Arellanez, Natanael Cervantes, Claudia E. Carrasco-Legleu, Gerardo Pavel Espino-Solis

**Affiliations:** 1Research Laboratories, Faculty of Physical Culture Sciences, Autonomous University of Chihuahua, Campus II, Periférico de la Juventud y Circuito Universitario S/N. Fracc, Campo Bello 31125, Mexico; esquintana@uach.mx (E.Q.-M.); jurodrig@uach.mx (J.M.R.-V.); ncervantes@uach.mx (N.C.); ccarrasco@uach.mx (C.E.C.-L.); 2Departamento de Ciencias Quimico Biologicas, Instituto de Ciencias Biomedicas, Universidad Autonoma de Ciudad Juarez, Av. Benjamín Franklin No. 4650, Zona Pronaf Condominio La Plata, Cd Juárez 32310, Mexico; argel.gastelum@gmail.com; 3National Laboratory of Flow Cytometry, Faculty of Medicine, and Biomedical Sciences, Autonomous University of Chihuahua, Circuito Universitario s/n, Campus II, Chihuahua 31125, Mexico

**Keywords:** Natural Killer cells, exercise, cytokine profile

## Abstract

Physical exercise generates a systemic response in the immune system. It has been observed that cell populations respond to exercise stimuli, especially Natural Killer cells, whose number increase within minutes of starting physical exertion. This study aimed to evaluate the acute effect of moderate- and high-intensity exercise on immunological markers in healthy women. As specific objectives, the percentages of CD3^-^CD56^+^ Natural Killer total cells, CD56^bright^CD16^dim^ effector subpopulation, CD56^dim^CD16^bright^ cytotoxic subpopulation, NKG2A inhibition receptor, NKG2D activation receptor, and NKT cells were analyzed. In addition, the levels of the cytokines IL-1β, IL-6, IL-8, IL-10, IL-12p70, and TNF and the chemokines CCL5/RANTES, CXCL9/MIG, CCL2/MCP-1, and CXCL10/IP-10 were also analyzed. Natural Killer total cells showed an increase in their percentage in both exercise protocols (*p* = 0.001 for the moderate-intensity group and *p* = 0.023 for the high-intensity group); however, only in the high-intensity exercise session was there an increase in the CD56^dim^CD16^bright^ cytotoxic subpopulation (*p* = 0.014), as well as a decrease in CD56^bright^CD16^dim^ effector subpopulation (*p* = 0.001) and their NKG2A inhibition receptor (*p* = 0.043). An increase in IL-6 was observed after the high-intensity exercise session (*p* = 0.025). Conclusions. Physical exercise influences immunological markers and shows an acute response to moderate- or high-intensity exercise.

## 1. Introduction

Physical exercise generates a systemic response in the immune system, in both the innate and adaptive immune system. This systemic response is reflected in the leukocyte populations and is characterized by an increase in peripheral blood and redistribution to organs and tissues, mainly Natural Killer cells (NK), followed by CD8^+^ T cells, B cells, and then CD4^+^ T cells. NK cells are the cells that are most sensitive to the effect of physical exercise, as their numbers increase within minutes of starting physical exertion [1,2]. These cells are part of the body’s first line of defense against pathogens or cancer cells, so establishing an adequate exercise load to increase them and improve their cytotoxic capacity can improve the prevention of various diseases, in addition to identifying the behavior of its subpopulations and its specific receptors [3,4]. The increase of NK cells in peripheral blood is due to the release of cytokines during muscle contraction which induce the proliferation, maturation, and activation of NK cells; another way that these cells increase is due to the catecholamines released by exercise, mobilizing epinephrine-dependent NK cells; finally, increased blood flow stimulates NK reservoirs located in the vascular endothelium. The main subpopulations of NK cells are CD56^bright^CD16^dim^ (effector subpopulation) with secretory function of cytokines and CD56^dim^CD16^bright^ (cytotoxic subpopulation), which have the function of eliminating target cells through perforins and granzymes; they correspond to approximately 10% and 90% in peripheral blood of NK cells, respectively [4].

On the other hand, cytokines and chemokines are the molecules responsible for communication and signaling between the components of the immune system and are involved in the acute and chronic response to exercise; they facilitate the distribution of lymphocytes, neutrophils, monocytes, and other cells, which participate in the elimination of the antigen and the recovery of homeostasis. Some of these soluble markers are secreted by contracting muscle fibers and may have anti- and pro-inflammatory properties [5].

The effect of the type, intensity, and duration of exercise has been reported on immunological markers [6,7]. It has been suggested that moderate-intensity exercise shows greater benefits to the immune system by stimulating NK cell activity, improving antigen presentation, reducing inflammation, and preventing cellular ageing. Instead, the high-intensity, long-duration exercise decreases the proliferation of T cells and their cytotoxic capacity [8], in addition to the inhibition of immunomodulatory cytokines, which can cause immunity to remain suppressed for hours or days after strenuous exercise, making the body susceptible to infections [9]. However, the effect of high-intensity exercise load at short intervals and the possible immune system stimulation without causing immunosuppression has not been reported, contrasting with long-duration intense exercise [1].

It is necessary to know the systemic immune response to an exercise load in different modalities in healthy people to establish the bases for more assertive exercise prescription in people with health conditions with a compromised immune system. The data currently accumulated on the effects of exercise in relation to the immune system show the knowledge gap that remains to be explored to take advantage of an adequate dosage of exercise as a complementary therapy to a comprehensive treatment. Our hypothesis is that physical exercise of moderate intensity and high intensity of short duration could contribute to the enhancement of the response of markers of the immune system for the prevention or treatment of diseases; based on this, this study aimed to evaluate the acute effect of moderate- and high-intensity exercise on immunological markers in healthy women. As specific objectives, the percentages of CD3^-^CD56^+^ Natural Killer total cells, CD56^bright^CD16^dim^ effector subpopulation, CD56^dim^CD16^bright^ cytotoxic subpopulation, NKG2A inhibition receptor, NKG2D activation receptor, and NKT cells were analyzed. In addition, the levels of the cytokines IL-1β, IL-6, IL-8, IL-10, IL-12p70, and TNF and the chemokines CCL5/RANTES, CXCL9/MIG, CCL2/MCP-1, and CXCL10/IP-10 were also analyzed.

## 2. Materials and Methods

### 2.1. Participants and Study Design

A quasi-experimental study was carried out to evaluate the acute effect of moderate- (*n* = 23) and high-intensity (*n* = 9) physical exercise on immunological markers by flow cytometry. The study included 30 women recruited through a call. Inclusion criteria included more than 18 years of age, no acute or chronic conditions, not taking beta-blocker drugs, and no pregnancy.

The project obtained the approval of the Research Ethics Committee of Hospital Angeles de Chihuahua (Registration: 18-CI-08-019-009), adherence to the standards of the Declaration of Helsinki was ensured, and each participant was asked to sign the respective letter of informed consent; a medical authorization was also required for the participants of the high-intensity session.

### 2.2. Physical Exercise Session

The moderate session consisted of 30 min on a treadmill at 40% of the reserve heart rate, supervised with Polar OH1 monitor (Model 2L WR30). The high-intensity session consisted of four minutes of intervals of eight cycles of 20 s of “All-Out” exercise (skipping, jump squat, butt kicks, and jumping jack) for every 10 s of active rest. Blood samples were taken before and immediately after each exercise session. All subjects completed the exercise session, and it was performed individually to ensure blood extraction with the same post-exercise timing.

### 2.3. Flow Cytometry

Blood samples were lysed with Red Blood Cell Lysing Buffer Hybrid-Max^TM^ (Sigma-Aldrich Co., LLC, REF R7757, San Louis, MO, USA) and stained with anti-CD3 PerCP (Becton Dickinson, #cat. 347344, clone SK7, CA, United States), anti-CD16 APC-H (Thermo Fisher Scientific, #cat. 560195, clone 3G8, Madrid, Spain), anti-CD56 APC (Becton Dickinson, #cat. 555518, clone B159, San Jose, CA, USA), anti-NKG2D FITC (Thermo Fisher Scientific #cat. 320820, clone 1D11, Walthman, MA, USA), anti-NKG2A Biotinylated (Miltenyi Biotec, #cat. 130113564, clone REA110, Bergisch Gladbach, Germany), and Streptavidin PE (Becton Dickinson, #cat. 349023, CA, United States). FMO (Fluorescence Minus One) controls were performed to identify fluorescence limits and establish populations and Gate Strategies (Figure 1).

Cytokine levels were analyzed by a Cytometric Bead Array, using Human Inflammatory Cytokines Kit BD^TM^ (# cat. 551811, Franklin Lakes, NJ, USA) for the quantification of IL-1β, IL-6, IL-8, IL-10, IL-12p70, and TNF, and the chemokine levels were analyzed with the Human Chemokine Kit BD^TM^ (#cat. 552990, NJ, United States) for the quantification of CCL5/RANTES, CXCL9/MIG, CCL2/MCP-1, and CXCL10/IP-10. In both protocols, 5 μL of each capture bead, 25 μL of the PE Detection Reagent, and 50 μL of plasma were prepared for each sample in 1.5 mL tubes; they were incubated for 1.5 h at room temperature, protected from light, and washed with 1 mL of Wash Buffer. Samples were then resuspended with 300 μL of Wash Buffer.

Concentrations in pg/mL were calculated from the construction of a standard curve for each cytokine and chemokine.

The Thermo Fisher Scientific Attune NxT flow cytometer and FlowJo X Software (Becton Dickinson ™ v10.7) were used for the analysis of NK cell populations, cytokines, and chemokines. For the quantification of cytokines and chemokines, the size (FWD) with the complexity (SSC) of the spheres was considered to delimit the fluorescence intensity corresponding to each one and calculate their concentration in picograms per milliliter (pg/mL).

### 2.4. Statistical Analysis

A data analysis was performed in the R 4.2.1 environment for statistical analysis [10]. The normality of the data and the homogeneity of variances were evaluated using the Shapiro–Wilk and Bartlett tests, respectively. The cosine transformation was applied to both experimental groups when significant deviations from normality were found. Student’s *t*-test for related samples was applied to compare before and after physical exercise, while a Student’s *t*-test for independent samples (α = 0.05) was used to compare the two physical exercise protocols in the total sample of each group. Moreover, categories were generated to analyze NK cell data with age categories (≤40 years vs. >40 years); Body Mass Index (normal weight vs. overweight and obesity) according to the Quetelet’s Index and the World Health Organization, which considers a normal weight to be 18.5–24.9 kg/m^2^, overweight to be ≥25 kg/m^2^, and obesity to be ≥30 kg/m^2^; and physical condition (sedentary vs. active) assessed by the International Physical Activity Questionnaire in short form (IPAQ-SF) and based on the WHO physical activity recommendations.

## 3. Results

Samples of 30 women who agreed to participate in the study were analyzed. Seven participants who were taking antidepressant, anxiolytic, or sleeping pills were excluded (due to possible interference with the immune response). The age of patients ranged between 21 and 55 years, and they had an average BMI of 25.5 kg/m^2^. These patients performed the moderate-intensity session, while the high-intensity session was completed by nine participants between the ages of 21 and 30.

The percentage of the total NK cell markers CD3^-^CD56^+^, the subpopulation effectors CD56^bright^CD16^dim^ and cytotoxic CD56^dim^CD16^bright^, the receptors NKG2A for inhibition and NKG2D for activation, and the percentage of total CD3^+^CD56^+^ NKT cells were evaluated.

For the moderate-intensity-session group, it was found that only in the population of total CD3^-^CD56^+^ NK cells was there a statistically significant increase from 5.7 ± 0.7% to 7.3 ± 0.8% (*p* = 0.001) (Figure 2). However, when considering the age groups, BMI, and physical condition, changes were observed in most of the markers of these cells (Table 1).

The high-intensity session was performed by nine of the participants; for these women, we found an increase in the population of total CD3^-^CD56^+^ NK cells (7.75 ± 1.5 vs. 19.04 ± 4.8, *p* = 0.023) and the CD56^dim^CD16^bright^ cytotoxic population (83.48 ± 7.7 vs. 87.26 ± 8.5, *p* = 0.014), as well as a decrease in the effector population CD56^bright^CD16^dim^ (5.27 ± 1.0 vs. 2.83 ± 1.2, *p* = 0.001) and the inhibition receptor NKG2A (35.02 ± 3.5 vs. 32.08 ± 3.4, *p* = 0.043). In this group, it was impossible to make comparisons according to age categories, BMI, or physical condition (Figure 2).

The comparison between the post-exercise results of the moderate-intensity vs. high-intensity exercise sessions showed a higher increase in the total CD3-CD56+ NK cells (7.42 ± 0.8 in contrast to 19.04 ± 2.6, *p* < 0.01) and a higher decrease in the effector population CD56^bright^CD16^dim^ (4.69 ± 0.4 against 2.83 ± 0.7, *p* = 0.024); a decrease in the inhibition receptor NKG2A (36.36 ± 2.6 in comparison 32.08 ± 3.4, *p* = 0.021) was also observed in those who performed high-intensity session (Figure 2).

The cytokine and chemokine profile analyses (Figure 3 and Figure 4) obtained from six participants of each exercise protocol were included. The results showed a slight increase in IL-6 (11.47 ± 0.2 vs. 12.12 ± 0.2, *p* = 0.025) after four minutes of high-intensity exercise.

## 4. Discussion

We analyzed 23 women between 21 and 55 years, with a mean of 37 ± 8.5 and a mean BMI of 25.5 ± 4.2, who performed a moderate-intensity exercise session. After this, nine participants performed a high-intensity session. The percentages of NK cells and their populations, as well as the cytokine and chemokine profile, were evaluated to determine the immune response to an exercise load.

### 4.1. NK Cells

Total CD3^-^CD56^+^ NK cells showed an increase in peripheral blood after moderate- and high-intensity physical exercise. The increase in the total population of NK cells agrees with some reports about the acute effect of exercise [11,12,13,14,15]. The reported results and those obtained in this work suggest that this response is based on the effect caused by the epinephrine released during physical exertion, which stimulates the beta-adrenergic receptors present in NK cells, as well as the effect of increased blood flow on NK reservoirs present in the vascular endothelium [4,16,17]. However, these authors only reported the percentage of total NK cells, without considering the behavior of their populations or their receptors; likewise, in their results, not all of them expressed whether there is a possible influence of individual factors such as age, BMI or physical condition.

In moderate-intensity exercise, the effector population decreased regardless of age, while the cytotoxic population increased only in those over 40 years of age, a response similar to that found by Millard et al. in 2013 [18]; however, Campbell et al. (2018) [2] mention that, despite this response, the cells have a lower cytotoxic capacity due to the effect of ageing. Sellami et al. (2018) [19] and Lutz et al. (2005) [20] concluded that ageing induces changes in the phenotype and function of NK cells, finding that these cells present a higher response in the adult population compared to young individuals. On the other hand, Krishnaraj et al. (1998) [21] showed that the increase in catecholamines released during exercise is greater in older individuals, and it directly influences the beta-adrenergic receptors of cytotoxic NK cells. It is important to highlight the low migratory potential of the cytotoxic population towards lymphoid tissues, remaining constant in peripheral blood, in contrast to the affinity of the effector population for secondary lymphoid tissues, where it is considered that its decrease due to exercise may be due to a possible relocation of effector NK cells to other tissues, thus increasing immunosurveillance in the body [2].

The effector population decreased regardless of BMI, while, only in overweight or obese participants, an increase in the cytotoxic population was observed. There is insufficient information on the phenotype of NK cells in overweight and obese subjects. However, Viel et al. (2016) [22] reported that the higher the BMI, the higher the percentage of NK cells; nevertheless, these increased cells in people with obesity presented impaired cytotoxic activity due to having a low degranulation capacity, which could contribute to a greater susceptibility to developing cancer or infectious diseases. No studies were found where the effect of exercise on NK cells in subjects with a high BMI was analyzed; however, Viel et al. (2016) [22] and Barh et al. (2018) [23] highlight the importance of exercise for the regulation of body weight since the alterations generated in NK cells, such as the decrease in degranulation capacity or low cytotoxic activity, seems to be corrected with weight loss. In addition, overweight patients could benefit from the increase of the cytotoxic population generated by the exercise. On the other hand, only in the participants with an active physical condition was it possible to observe changes in the NK cell populations, the decrease in the effector population, and the increase in the cytotoxic population. These results suggest a direct relationship with a previous report [2] about the effect that regular physical activity can have on limiting or delaying the ageing of the immune system and can explain the reduction in the incidence of cancer among people who are physically active throughout life [24].

In women older than 40 years, a slight but significant decrease in NKG2A receptor inhibition was observed; according to previous studies, the effector population of NK cells has a high amount of the NKG2A receptor compared to the cytotoxic population, and this could indirectly explain the decrease in this marker. Additionally, the decrease in the percentage of this receptor indicates that the NK cell has a more mature stage and, consequently, a more cytotoxic phenotype. On the other hand, a slight decrease in receptor activation NKG2D was observed in the mild-intensity-group participants with average weight, which could reflect a security measure to prevent excessive stimulation of NK cells that would lead to immunopathology [25]. Previous reports have studied the response of this receptor to physical exercise. There were no statistical differences [18,26]; however, it is not possible to establish a comparison between results due to the assortment of samples and intensity of exercise.

In the high-intensity session, an increase in the total and cytotoxic population of NK cells was observed, and a decrease in the effector population and the NKG2A receptor in all participants (*n* = 9) was observed compared to the moderate-intensity session. This could indicate that the greater the stimulus, the greater the NK cell response is observed, as long as it is not a prolonged workload, since it has been found that, during exercises that combine high intensity and long duration, immunosuppression occurs [27].

### 4.2. Cytokine and Chemokine Profile

Circulating cytokines and chemokines are essential for the communication of signals from the immune system; however, the secretion of these factors can be altered by various stimuli throughout life, including physical exercise [28,29]. Moderate-intensity exercise has shown anti-inflammatory effects in the context of low-grade chronic inflammation. Nevertheless, high-intensity interval training has begun to be studied for health purposes due to its ability to obtain cardiometabolic benefits with shorter exercise sessions; however, studies under this modality are still scarce compared to those carried out regarding continuous moderate exercise [30].

In the present study, a slight increase in the concentration of IL-6 was observed after a short session of high-intensity exercise. This cytokine has been studied for its pro- and anti-inflammatory effects and its relationship with physical exertion. Strenuous exercise can increase the plasma IL-6 levels in healthy individuals in a single session due to muscle damage [31]; this increase in peripheral blood has an influence on the reduction of concentrations of postprandial glucose and reduced insulin secretion, in addition to playing a role in appetite regulation and visceral fat mass loss [32].

In addition, in recent years, it has been observed in murine models that NK cells can suppress tumor cell growth through the mobilization of epinephrine- and IL-6-dependent cytotoxic NK cells secreted during exercise [32,33,34].

The acute effect of exercise generates a complex response of pro- and anti-inflammatory events that depends on the type, intensity, duration, and specific conditions of each individual, such as the age, body composition, physical condition, or presence of diseases. The practice of regular exercise has been considered an anti-inflammatory therapy. Once the immediate inflammatory response to physical exertion is reestablished, where it has been reported that it returns to its basal level from 5 to 24 h after exercise, a response that seems to be necessary for tissue repair [35].

It is important to highlight that the participants in this investigation were women, which indicates that many variables would need to be monitored or evaluated to confirm that the changes found in our study and other investigations are exclusively because of exercise since the female hormonal variations or the menstrual-cycle stage could influence the immunological markers evaluated. The immune response, both cellular and humoral, can be modified according to the different phases of the menstrual cycle. In the follicular phase, the cellular immune response predominates. During the preovulatory time, there is a decrease in the cytolytic activity of NK cells, while in the luteal period, there is a change in the immune response from cellular to humoral [36,37]. In the present study, the menstrual-cycle phase of the participants was not considered.

As limitations, the number of surface markers related to NK cells needs to be redesigned, enlarged, and improved to be able to particularly determine their phenotype and the effect of exercise on the cytotoxic activity; in addition, a functional assay such as antibody-dependent cellular cytotoxicity (ADCC) could be performed as an indicator to monitor if exercise protocols are capable of triggering the effector capabilities of NK cells. Furthermore, we mentioned the need to increase the number of subjects evaluated in the cytokine and chemokine analyses, even though valuable data were obtained from a small number of samples. However, the menstrual-cycle status of participants is a critical experimental variable to avoid a hormonal-bias impact on the results. Thus, a kinetic-experiment time recovery and their impact on cytokine and chemokine concentrations must be performed to find the appropriate interrogation time for these particular analytes. It is necessary to generate studies to assay the acute and chronic effects of a specific physical activity in concert with the follow-up at different times compared with other exercise methodologies and other populations at risk, as well as recruit a homogeneous sample of patients for the study groups.

## 5. Conclusions

Cell population markers and soluble factors are a fundamental part of understanding the systemic response generated by exercise on the immune system. This study showed the acute effect of physical activity on the percentage of total CD3^-^CD56^+^ NK cells; their two main effector subpopulations, CD56^bright^CD16^dim^ and cytotoxic CD56^dim^CD16^bright^; and their inhibition NKG2A and activation NKG2D receptors, as well as the effect on concentrations of the IL-1β, IL-6, IL-8, IL-10, IL-12p70, and TNF cytokines and on the CCL5/RANTES, CXCL9/MIG, CCL2/MCP-1, and CXCL10/IP-10 chemokines. These populations respond to moderate and high exercise intensities, and there is an influence of the individual characteristics of the participants, such as age, BMI, and physical condition (in the case of NK cells).

Although more evident responses have been observed with high-intensity exercise, the intention of evaluating the behavior of NK cell populations before a session of continuous aerobic exercise at a moderate intensity arises from the need to be able to implement these results in vulnerable populations that need exercise protocols that are suitable for their physical condition and health, such as older adults, sedentary people, and people with chronic diseases such as cancer or viral diseases.

## Figures and Tables

**Figure 1 sports-11-00189-f001:**
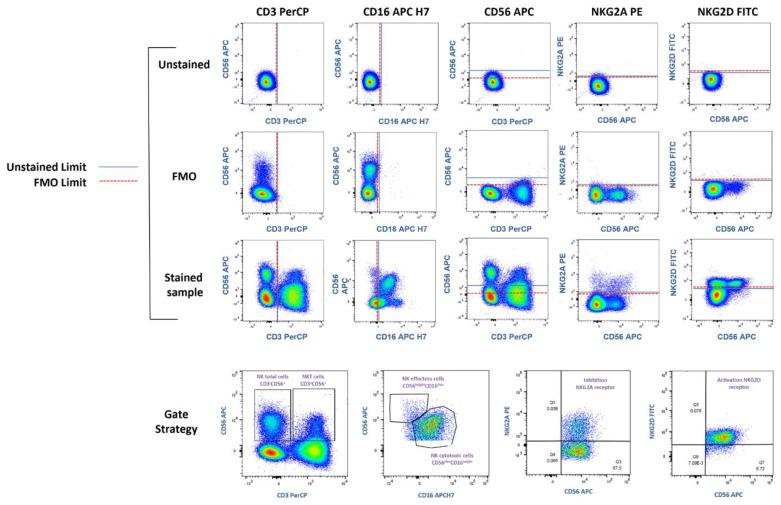
Fluorescence Minus One (FMO) and Gate Strategy. Representation of the Fluorescence control Minus One. The dot plot of the unstained control, FMO control, and fully stained sample is shown for each fluorochrome. The blue line represents the limit of the unstained sample, and the red line the FMO limit. The Gate Strategy shows the analysis of NK total cells, their NK effectors CD56^bright^CD16^dim^ and NK cytotoxic CD56^dim^CD16^bright^ subpopulations, and their NKG2A and NKG2D receptors, as well as NKT cells CD3^+^CD56^+^. Subpopulations of NK cells were derived from the NK total population CD3^-^CD56^+^.

**Figure 2 sports-11-00189-f002:**
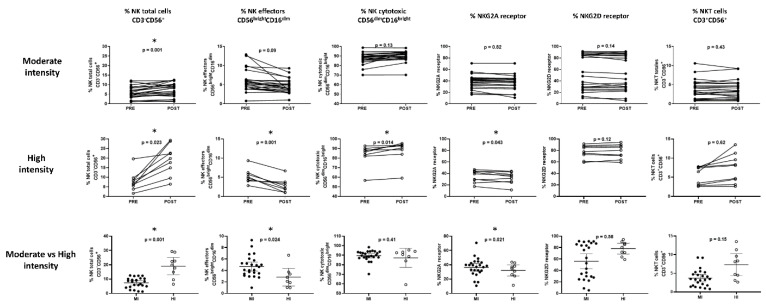
Acute response of the Natural Killer sub-populationto a single load of physical exercise at moderate and high intensity and the difference in response between both intensities Paired Student’s *t*-test for related samples and Student’s *t*-test for independent samples. * significant differences *p*-value < 0.05.

**Figure 3 sports-11-00189-f003:**
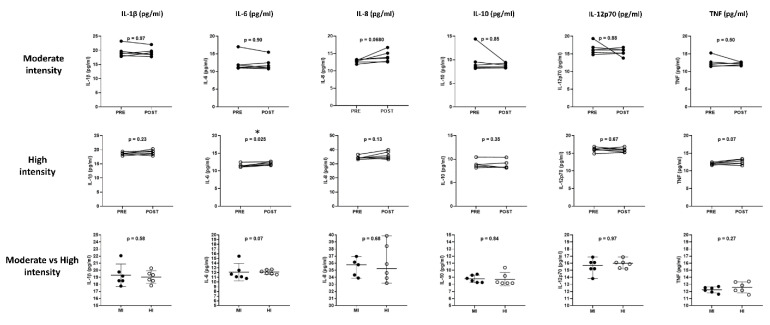
Acute response of the cytokine profile to a single load of physical exercise at moderate and high intensity and the difference in response between both intensities. Paired Student’s *t*-test for related samples and Student’s *t*-test for independent samples. * significant differences *p*-value < 0.05.

**Figure 4 sports-11-00189-f004:**
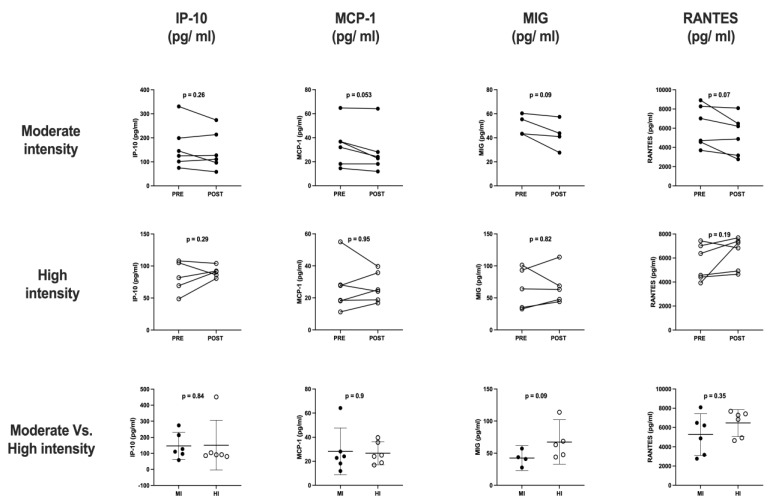
Acute response of the chemokines profile to a single load of physical exercise at moderate and high intensity and the difference in response between both intensities Paired Student’s *t*-test for related samples and Student’s *t*-test for independent samples.

**Table 1 sports-11-00189-t001:** Pre–post exercise values on NK cell markers according to age, BMI, and physical condition for moderate-intensity group.

	Age				BMI				Physical Condition			
	≤40 Years (*n* = 16)		>40 Years (*n* = 7)		Normal Weight (*n* = 14)		Overweight and Obesity (*n* = 9)		Sedentary (*n* = 11)		Active (*n* = 12)	
Cell Population (%)	Pre	Post	Pre	Post	Pre	Post	Pre	Post	Pre	Post	Pre	Post
NK total cells CD3^-^CD56^+^	5.8 ± 0.9	7.2 ± 1.0 *	5.4 ± 0.7	7.5 ± 1.0 **	6.0 ± 0.9	7.3 ± 1.0	5.1 ± 0.8	7.2 ± 1.10 **	5.6 ± 1.2	6.7 ± 1.2	5.8 ± 0.7	7.9 ± 1.0 **
NK effectors CD56^bright^CD16^dim^	5.4 ± 0.7	4.3 ± 0.5 *ᵻ	7.5 ± 1.1	5.5 ± 0.8 *	5.7 ± 0.5	5.1 ± 0.5 *	6.6 ± 1.4	4.1 ± 0.8 *	6.6 ± 1.0	4.8 ± 0.7 ᵻ	5.5 ± 0.7	4.6 ± 0.5 **
NK cytotoxic CD56^dim^CD16^bright^	88.4 ± 1.60	90.2 ± 1.5	86.5 ± 2.1	89.1 ± 2.1 *	87.9 ± 1.5	88.7 ± 1.6	87.7 ± 1.3	91.6 ± 1.1	86.8 ± 2.4	89.3 ± 2.2	88.8 ± 1.2	90.4 ± 1.0 *
NKG2A receptor	35.4 ± 2.6	33.2 ± 2.8 ᵻ	45.5 ± 4.8	44.5 ± 5.0 *	41.3 ± 3.5	40.0 ± 3.4 ᵻ	34.1 ± 3.0	31.4 ± 3.5 ᵻ	32.6 ± 3.0	30.4 ± 3.7 ᵻ	43.9 ± 3.2	42.4 ± 3.2 ᵻ
NKG2D receptor	66.1 ± 7.4	63.8 ± 7.6 ᵻ	36.5 ± 9.1	37.9 ± 9.8	60.5 ± 7.7	58.4 ± 7.8 *	51.8 ± 12.0	51.9 ± 12.0	52.7 ± 10.0	52.7 ± 10.4	61.1 ± 8.4	58.8 ± 8.4
NKT cells CD3^+^CD56^+^	3.3 ± 0.9	3.2 ± 1.0	4.9 ± 1.3	4.8 ± 1.2	4.4 ± 0.7	4.3 ± 0.8	2.9 ± 0.4	2.7 ± 0.4	4.1 ± 0.8	3.8 ± 0.7	3.5 ± 0.7	3.6 ± 0.7

All data are represented as mean ± standard error; * *p* < 0.05, and ** *p* < 0.01 vs. pre; ᵻ, *p*-value obtained by means of cosine data transformation; BMI, Body Mass Index.

## Data Availability

The data used to support the findings of this study are available from the corresponding author upon request.

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
