# Peer review of "The Effect of Acute Physical Exercise on Natural Killer Cells Populations and Cytokine Levels in Healthy Women"

_sports, 2023, doi:10.3390/sports11100189_

Round 1

Reviewer 1 Report

Dear authors,

About your paper titled: The effect of acute physical exercise on Natural Killer cells populations and cytokine levels in women.

First, thank you for the opportunity to review this work. Your words had a highly practical application to the use of physical exercise as medicine against disease. In this way, I have some suggestions as follows in my reviews:

Title

I suggest you add the health state of the women, i.e., healthy women… to better describe the population which were studied.

Introduction

However, the effect of high-intensity exercise load at short intervals and the possible immune system stimulation without causing immuno-suppression has not been reported.

In which population it has been reported? You need to clarify it in your text because it could be studied in other populations, so let it be clear.

If there were previous similar studies conducted within other populations, you could describe them to expose the state-of-art knowledge. In addition, add your hypothesis right after this topic, right before your objectives.

Physical exercise session

The moderate session consisted of 30 minutes on a treadmill at 40% of the reserve heart rate monitor with a heart rate monitor. Add the monitor’s basic information.

Results

Improve the figure’s 1 resolution.

Make the same for figure 2.

Table 1 needs to be better organized. Please, make it.

Improve figure 2 quality also.

Discussion

The discussion was quite good. I only suggest you carefully revise the end of sentences and question if it is necessary to add references after some discussions.

Conclusion

You need shortly the conclusion. In addition, I suggest you rearrange your limitations and strengths for the last discussion’s paragraph or even create a topic of strengths and limitations and future directions.

Kind regards, 

 The English is well-written and only need a slight review.

Author Response

Sep 16, 2023

Dear Editor

MDPI - Sports

We are pleased to submit for your consideration our revised manuscript entitled, “The effect of acute physical exercise on Natural Killer cells populations and cytokine levels in women”. Manuscript ID: Sports-2568551. We appreciate the time and effort that you and the reviewers have dedicated for providing your valuable feedback on this manuscript. We are grateful to the reviewers for their valuable comments on our paper. We have been able to incorporate changes to reflect most of the suggestions provided by the reviewers. We have highlighted the changes in yellow within the manuscript.

You will find here point-by-point responses to the reviewers’ comments and concerns.

Gerardo Espino, PhD

Reviewer 2 Report

It is known that physical exercise generates a systemic response in the immune system. Literature data provide numerous examples of changes in different immune cells upon physical exercise. This manuscript was aimed at evaluating the acute effect of moderate and high-intensity exercise on immunological markers in women. It is an interesting work. However, I have some concerns I’d like to express.

·         There is a typographic mistake in few sections of the manuscript (Abstract, Introduction, Conclusions). There should be IL-1ß instead of IL-ß.

·         To give the reader broader insight into the topic, at least of few sentences regarding the influence of physical exercise on Th (CD3+CD4+) and Tc (CD3+CD8+) cells should be provided.

·         Participants of the study were women. What was their hormonal (menstrual cycle status) in the time of the study? Could it have an impact on the results? Please, discuss shortly.

·         The Authors declare that blood samples were taken immediately after the exercise session. However, it is known that the changes in immune cells are within minutes after the exercise, the Authors should be more precise. Was the exercise session at the same time for all participants? Did they complete the exercise at exact the same time? It seems important in the context of the time between completing the exercise and blood taking.

·         Also, I miss the data in recovery time, especially in regard of cytokine/chemokine concentrations. They change in a longer time, so it would be important to know the pattern of their changes.

·         Describing equipment, reagents full data of the manufacturer (name, city, country) should be provided at first mention. Additionally, providing data of antibodies used in the study, a clone should be mentioned.

·         Did the Authors use any software for calculating cytokine/chemokine levels? Please, describe.

·         There is a need to clarify why only participants up to 30 years of age performed high-intensity sessions. It could influence the results and is inconsistent with the division into age groups.

·         What were the criteria of division participants into BMI and physical condition groups? The latter requires explanation – what kind/time of activity decided to put a participant into sedentary/active group.

·         It is also important why cytokine/chemokine levels from only 6 participants were analyzed? How were they chosen?

·         Did the Authors consider the fact that adipose tissue generates inflammation? It is important, since pro- and anti-inflammatory factors were analyzed.

The references section misses references 30-34 that are mentioned in the text of the manuscript. Please, complete.

Author Response

Sep 16, 2023

Dear Editor

MDPI - Sports

We are pleased to submit for your consideration our revised manuscript entitled, “The effect of acute physical exercise on Natural Killer cells populations and cytokine levels in women”. Manuscript ID: Sports-2568551. We appreciate the time and effort that you and the reviewers have dedicated for providing your valuable feedback on this manuscript. We are grateful to the reviewers for their valuable comments on our paper. We have been able to incorporate changes to reflect most of the suggestions provided by the reviewers. We have highlighted the changes in yellow within the manuscript.

You will find here point-by-point responses to the reviewers’ comments and concerns.

Gerardo Espino, PhD

Universidad Autonoma de Chihuahua

Round 2

Reviewer 2 Report

The Authors made suggested improvements to the manuscript.

There are a few typo mistakes (e.g. parentheses instead of brackets when citing references) left to be eliminated at the proof stage.